# Diabetic Macular Edema: Current Understanding, Molecular Mechanisms and Therapeutic Implications

**DOI:** 10.3390/cells11213362

**Published:** 2022-10-25

**Authors:** Jingfa Zhang, Jingxiang Zhang, Chaoyang Zhang, Jingting Zhang, Limin Gu, Dawei Luo, Qinghua Qiu

**Affiliations:** 1Department of Ophthalmology, Shanghai General Hospital (Shanghai First People’s Hospital), School of Medicine, Shanghai Jiao Tong University, 100 Haining Road, Hongkou District, Shanghai 200080, China; 2National Clinical Research Center for Eye Diseases, Shanghai 200080, China; 3Shanghai Key Laboratory of Ocular Fundus Diseases, Shanghai 200080, China; 4Shanghai Engineering Center for Visual Science and Photomedicine, Shanghai 200080, China; 5Shanghai Engineering Center for Precise Diagnosis and Treatment of Eye Diseases, Shanghai 200080, China; 6Nursing Department, People’s Hospital of Huangdao District, Qingdao 266400, China; 7Department of Ophthalmology, Shanghai Aier Eye Hospital, Shanghai 200336, China

**Keywords:** diabetic retinopathy, diabetic macular edema, blood-retinal barrier breakdown, drainage dysfunction, inflammation, anti-VEGF, proteomics, metabolomics

## Abstract

Diabetic retinopathy (DR), with increasing incidence, is the major cause of vision loss and blindness worldwide in working-age adults. Diabetic macular edema (DME) remains the main cause of vision impairment in diabetic patients, with its pathogenesis still not completely elucidated. Vascular endothelial growth factor (VEGF) plays a pivotal role in the pathogenesis of DR and DME. Currently, intravitreal injection of anti-VEGF agents remains as the first-line therapy in DME treatment due to the superior anatomic and functional outcomes. However, some patients do not respond satisfactorily to anti-VEGF injections. More than 30% patients still exist with persistent DME even after regular intravitreal injection for at least 4 injections within 24 weeks, suggesting other pathogenic factors, beyond VEGF, might contribute to the pathogenesis of DME. Recent advances showed nearly all the retinal cells are involved in DR and DME, including breakdown of blood-retinal barrier (BRB), drainage dysfunction of Müller glia and retinal pigment epithelium (RPE), involvement of inflammation, oxidative stress, and neurodegeneration, all complicating the pathogenesis of DME. The profound understanding of the changes in proteomics and metabolomics helps improve the elucidation of the pathogenesis of DR and DME and leads to the identification of novel targets, biomarkers and potential therapeutic strategies for DME treatment. The present review aimed to summarize the current understanding of DME, the involved molecular mechanisms, and the changes in proteomics and metabolomics, thus to propose the potential therapeutic recommendations for personalized treatment of DME.

## 1. Overview of Diabetic Retinopathy (DR) and Diabetic Macular Edema (DME)

Diabetes mellitus is a chronic, metabolic disease, which is characterized by a prolonged period of hyperglycemia. According to the reports of International Diabetes Federation (IDF) Diabetes Atlas 10th edition (https://diabetesatlas.org, accessed on 21 October 2022), the prevalence of diabetes has continued to increase globally, and diabetes still remains a significant challenge to the health and well-being of people. As estimated in IDF Diabetes Atlas 10th edition, there were 537 million adults (20–79 years) with diabetes in 2021. This number is estimated to rise to 643 million by 2030 and 783 million by 2045. 

Given the global pandemic of diabetes, diabetes causes serious damage to many organs, including the heart, blood vessels, eyes, and kidneys, which remains a major cause of heart attacks, stroke, blindness, and kidney failure (https://www.who.int, accessed on 21 October 2022). As one of the common complications of diabetes, diabetic retinopathy (DR) remains a major cause of visual impairment and blindness in working-age adults. Despite the advances in optimal control of systemic risk factors, i.e., hyperglycemia, hypertension and hyperglycemia, and the application of anti-vascular endothelial growth factor (VEGF) agents (Anti-VEGF), the prevalence of DR remains high in diabetic patients. Among diabetic patients, about one third of patients suffered from DR, which increases markedly after the age of 60 years due to the longer duration of diabetes [1]. A comprehensive systematic review and meta-analysis showed that the estimated pooled prevalence of DR was 28.41% (95% CI: 25.98 to 30.84), 25.93% (95% CI: 23.54 to 28.31), and 28.95% (95% CI: 26.57 to 31.32) in the diabetic population in general, in female, and in male, respectively; with no inter-gender difference for DR prevalence [1]. 

Diabetic macular edema (DME) represents the major cause of vision impairment in diabetic patients with an increasing prevalence worldwide [2]. DME is considered as the retinal thickening, which involves or approaches the fovea due to abnormal accumulation of fluid in the macula under diabetic condition. In diabetic retina, fluid accumulation in the macular area leads to increased central retinal/macular thickness, resulting in DME. The prevalence of DME varies widely, ranging from 4.2% to 14.3% in people with type 1 diabetes mellitus (T1DM) and 1.4% to 5.57% in people with type 2 diabetes mellitus (T2DM) [3,4]. Based on the findings of optical coherence tomography (OCT)-diagnosed DME, a systematic review showed the analysis of the pooled prevalence of DME was 5.47% (95% CI: 3.66–7.62%) overall worldwide, 5.81% (95% CI: 0.07–18.51%) in low-to-middle-income countries, and 5.14% (95% CI: 3.44–7.15%) in high-income countries, respectively [5]. The high prevalence of DR and DME not only seriously affects people’s life quality, but also lays a heavy economic burden on health care budgets.

## 2. OCT-Based DME Classification and Evaluation

Nowadays, OCT is widely used for the early detection of DME. Based on OCT findings, several types of DME were classified, i.e., diffuse edema (sponge-like diffuse retinal thickening), cystoid edema (retinal thickening with intraretinal cystoid change), macular edema with serous retinal detachment (macular thickening with subretinal fluid (SRF)), and mixed edema with at least two different edema types [6]. Other types of DME detected with OCT were also reported and proposed, including diffuse edema, cystoid edema, and cystoid degeneration, combined with different morphological features such as serous macular detachment, vitreomacular interface abnormalities, hard exudates, and photoreceptor status [7]. Arf S, et al. pointed that serous macular detachment should not be classified as a type of DME, which was an accompanying morphological change [7]. The author also indicated that cystoid macular degeneration should be differentiated from cystoid macular edema, because cystoid macular degeneration was correlated with the poor outcomes of the retinal function and morphology [7]. Based on OCT classification, cystoid macular edema is considered as intraretinal cystoid-like, hypo-reflective spaces with highly reflective septa separating the cystoid-like spaces in OCT B-scan, while cystoid macular degeneration is accepted when the horizontal diameter of the cystoid spaces larger than 600 μm (≥600 μm) [7], even though there is no consensus on its definition. 

In a retrospective study, DME was classified based on the localization and area of the fluid using en face images of swept-source OCT [8]. The retina was segmented into two segments, i.e., Segment 1, mainly comprising the inner nuclear layer (INL) and outer plexiform layer (OPL), including Henle’s fiber layer (HFL); and Segment 2, mainly comprising the outer nuclear layer (ONL). Fujiwara et al. reported that the DME patients with diffuse fluid at Segment 2 demonstrated significantly poorer visual acuity, higher disruption rates of ellipsoid zone (EZ), and greater central subfield thickness (CSFT) than those without fluid at Segment 2 [8]. The study indicated the impact of the localization and extent of the fluid on the visual outcome, and also suggested other biomarkers besides CSFT should be considered for evaluating the outcome of DME [8].

DME development and treatment outcome can also be well evaluated with OCT. In one study, 205 eyes from 129 diabetic patients underwent OCT angiography (OCTA) examination and were followed-up for at least 2 years [9]. The findings showed that some OCTA metrics could predict the progression of DR, which included foveal avascular zone area, vessel density, and fractal dimension of deep capillary plexus; whereas vessel density of superficial capillary plexus predicted the development of DME [9]. Toto L, et al. used widefield optical coherence tomography angiography (WFOCTA) to evaluate the changes in retinal capillary nonperfusion areas and retinal capillary vessel density in DME patients treated with an intravitreal dexamethasone implant [10]. The results showed that a reduction in retinal capillary nonperfusion areas after intravitreal injection of dexamethasone implant, suggesting the reperfusion of retinal capillaries might be drug-related in the early period [10].

## 3. The Pathogenesis of DME

DME is due to an imbalance between fluid entry, fluid exit and retinal hydraulic conductivity, leading to the accumulation of intraretinal fluid (IRF) or SRF [2]. IRF is the fluid accumulates in retinal parenchyma, mainly in the extracellular spaces of INL, OPL and ONL, while SRF is the fluid accumulation in subretinal space right underneath the neurosensory retina and above retinal pigment epithelium (RPE). Based on the Starling equation, in normal retina, the balance of influx and efflux of the fluid in retina is maintained by the blood-retinal barrier (BRB) integrity and the active drainage function of Müller glia and RPE [2,11]. The intact BRB and the active drainage function of both Müller glia and RPE maintain the retina under a relative dehydrated condition and normal function [2]. Under physiologic conditions, Müller glia removes the fluid from the retinal interstitial tissue to the blood vessels or vitreous, while RPE removes the SRF to the choroid by active transport [12,13]. However, the pathogenesis of DME is really complex, and it has still not been fully elucidated yet. Among the multiple, intricate mechanisms (Figure 1), DME develops mainly due to the two major underlying mechanisms, i.e., BRB breakdown, increasing fluid influx into retina parenchyma, and the decrease in drainage functions by Müller glia and RPE, resulting in the reduced fluid efflux out of retina [14,15]. Moreover, inflammation also plays a contributory role in BRB breakdown (Figure 1), resulting in DME [2,16,17,18].

The pathogenesis of DME starts with retinal hypoxia, resulting in the hyperpermeability of retinal capillary and increased intravascular pressure due to dysfunction of vascular autoregulation. The capillary hyperpermeability is mainly attributed to hypoxia-induced upregulation of VEGF, which plays a major role in the pathogenesis of DME. Currently, intravitreal anti-VEGF injection has become the first-line therapy in DME treatment, due to its many clinical benefits including functional and morphological improvements, and even halted or reversed DR progression. However, some patients respond unsatisfactorily to anti-VEGF injections. Based on the findings of Diabetic Retinopathy Clinical Research (DRCR) Retina Network Protocol T, there are 31.6~65.6% patients with DME who still exist with persistent edema even after regular intravitreal injection for at least 4 injections within 24 weeks [19], suggesting other pathogenic factors, beyond VEGF, might contribute to the pathogenesis of DME.

Moreover, it remains largely unknown whether or not these anti-VEGF agents can restore the drainage functions of Müller glia and RPE to facilitate the absorbance and transport of excess fluid out of retina. Alongside the VEGF pathway, DME formation is also attributed to other factors including leakage of microaneurysms, tractional effects by epiretinal membranes or posterior vitreous cortex, and inflammation from both retina and vitreous pool [2,17,18].

### 3.1. Two Forms of Edema in DME: Intracellular Edema and Extracellular Edema

In addition to the anatomic location of macular fluid accumulation as detected with OCT (SRF and IRF), there are two types of edema, i.e., intracellular edema and extracellular edema [17]. The fluid accumulation in the intracellular space is considered as intracellular swelling or cytotoxic edema, while fluid accumulation in the extracellular space is considered as extracellular edema or vasogenic edema [17]. 

In DME patients, both intracellular edema and extracellular edema can occur [2,17]. Diabetic intracellular edema can result from intracellular accumulation of sorbitol, lactate, and phosphates secondary to hyperglycemia-induced metabolic abnormalities [17]. Intracellular edema in the form of DME is the consequence of metabolic disturbances, which can be detected in RPE cells, Müller glia, and retinal neurons [20,21]. With the development of DME, the intracellular edema of the involved cells results in neuronal toxicity, contributing to vision loss and extracellular fluid volume increase [2]. Extracellular edema is mainly due to BRB breakdown, which leads to increased fluid accumulation and the decreased clearance of accumulated fluid with diabetes progression. Thus, in DME patients, intracellular edema, extracellular edema and even mixed edema can occur with the development of diabetes.

### 3.2. Drainage Dysfunction of Müller Cells Leading to Intracellular Edema

Müller glia, the major macroglia unique to the retina, transport and remove excess fluid from retinal parenchyma into the vitreous and retinal vessels [2,22,23]. Müller cells have many aquaporins, ion channels, transmembrane proteins, and enzymes. The numerous functions of Müller glia include drainage of extracellular fluid into the retinal vessels or vitreous body, regulation of retinal blood flow, maintenance of retinal pH by ionic homeostasis, their great conductance for potassium, glutamate recycling due to neuronal transmission and maintenance of glucose metabolism. 

In DR, the metabolism of Müller cells is disturbed, which compromises the drainage function of Müller cells and leads to intracellular accumulation of fluid and inadequate discharge of the fluid into retinal blood vessels or vitreous. Consequently, the intracellular swelling or cytotoxic edema of Müller cells leads to rupture of cell membranes as well as increased liquid accumulation in the extracellular space, presenting intraretinal cystoid abnormal spaces visible as focal defects in the deep capillary plexus and hypo-reflective cystoid spaces in B-scan of OCTA [2]. 

Müller cells maintain the homeostasis of ion and water in the retina via inward rectifying potassium channel 4.1 (Kir4.1) and aquaporin 4 (AQP4) [12,24]. The proteins Kir4.1 and AQP4 are anchored by Dystrophin 71 (Dp71) on Müller cellular membranes [25,26]. The downregulation or redistribution of Kir4.1 and AQP4 results in the intracellular edema of Müller cells, which was reported in many retinal diseases, including retinal vein occlusion, ischemia-reperfusion injury and DR [27,28,29]. In 3-month rat experimental DR, Kir4.1 was absent in the inner limiting membrane (ILM) and perivascular areas [30]. In our previous study using 3-month diabetic rat retinas, the expression of Kir4.1 and AQP4 were reduced [29]. The decreased expression and the altered distribution of Kir4.1 and AQP4 may be a molecular marker of the drainage dysfunction of Müller cells, leading to Müller intracellular edema. 

According to the published method [31], we observed the swollen apical processes of Müller glia as ribbon-like transparent gaps in ONL in semithin section of retina, indicating intracellular edema of Müller glia in streptozotocin-induced diabetic rat [29]. In our previous clinical observation in diabetic patients with DME, there was a strong positive correlation between INL thickness, where somas of Müller cells are located, and the CSFT, indicating that the Müller intracellular edema might contribute to DME development [32]. The Müller glia intracellular edema was also observed and reported in patients with DR when examined with OCTA, as indicated by the hypo-reflective cystoid edema spaces in the deep capillary network in both b-scan and en face of OCTA [2,29,33].

However, there is no direct evidence showing intracellular edema of RPE cells, which merits further study with high resolution multi-modal imaging to validate the intracellular edema in vivo both experimentally and clinically. With the advancement of the imaging technologies, metabolic imaging is promising to evaluate the function of RPE cells, which measures the intracellular levels of the metabolites non-invasively and real-timely [34]. Bianchetti G, et al. investigated the effect of docosahexaenoic acid (DHA) on the redox homeostasis in the human retinal pigment epithelial cell line (ARPE-19) under high-glucose conditions, using both metabolic imaging and molecular biology [34]. The metabolic imaging, using two-photon microscopy, showed that DHA treatment could increase intracellular nicotinamide adenine dinucleotide plus hydrogen (NADH) to upregulate the production of reductive species in high-glucose + DHA group compared with high-glucose treated cells [34]. Moreover, artificial intelligence-based metabolic imaging analysis will largely facilitate the evaluation of the metabolic changes in retinal cells, including RPE, both in vivo and in vitro [35,36].

### 3.3. BRB Breakdown Leading to Increased Fluid Leakage into Retina

The BRB protects retinal neuronal functions via regulating the protein, ion, and water to maintain an adequate microenvironment in retina. The BRB is consisted of both inner BRB and outer BRB. Abnormal fluid accumulation in the retina is mainly attributed to the BRB breakdown. Following BRB breakdown, the equilibrium is lost between hydrostatic and oncotic pressure gradients across the BRB leading to further progression of the macular edema [37,38]. In DR, BRB breakdown was observed both experimentally and clinically [2,37,39,40]. The loss of protective function of BRB is mainly due to the impairment of cells comprising the BRB and cell-cell junctions. Three BRB pathologies in DR are critical in the development of DME, i.e., pericyte loss, impaired cell-cell junctions, and capillary basement membrane thickening [41]. In DR, cell-cell junctions between the endothelial cells are lost, resulting in the increased leakage of plasma, lipid, and even red blood cells [42,43], manifesting clinically as edema, hard exudates and intraretinal hemorrhages. Bianchetti G, et al. analyzed and compared the alteration of erythrocyte membrane fluidity by calculating the generalized polarization (GP), representative for membrane fluidity of red blood cells, in T1DM patients with or without DR, and found that erythrocyte membranes of DR patients were more fluid than those of T1DM without DR, indicating alteration of erythrocyte membrane fluidity could represent a biomarker for DR development in T1DM patients [44]. The changes in GP in DR might be related to the early stages of diabetic microangiopathy, including capillary occlusion, microaneurysm formation, and retinal ischemia, resulting in BRB breakdown.

Hyperglycemia-induced upregulation of many cytokines, chemokines, and enzymes, such as angiopoietins, interleukins (ILs), cyclooxygenase-2 (COX-2), inducible nitric oxide synthase (iNOS), and matric metalloproteinases (MMPs), play a synergic role leading to the breakdown of BRB [41,45]. The predominant molecular mechanisms leading to BRB breakdown and subsequent DME include activated VEGF downstream pathways and inflammation leading to the increased production of many cytokines and growth factors [46,47], playing a causative role in BRB breakdown through multiple signaling pathways [46]. For example, tumor necrosis factor α (TNF-α) increases the expression of intercellular adhesion molecule 1 (ICAM-1), promoting leukocyte adhesion and leukostasis in early DR. TNF-α inhibition might provide a therapeutic target to prevent BRB breakdown, retinal leukostasis, and apoptosis in DR [46,48]. Interleukin-1β (IL-1β) stimulates reactive oxygen species (ROS) production and accelerates the apoptosis of retinal capillary endothelial cells through activating the nuclear factor kappa B (NF-κB) pathway under diabetic conditions. [49,50]. Furthermore, the activated kallikrein-kinin system in DR also contributes to retinal edema, hemorrhage and neovascularization through activation of bradykinin B1 and B2 receptors, resulting in vasodilation, vascular permeability, inflammation, and leukostasis [51]. 

Alongside inner BRB breakdown, outer BRB breakdown can also participate the pathogenesis of DME. In experimental DR, the degeneration of RPE and the breakdown of outer BRB were observed [39,40]. In early diabetic rats, the majority leakage of fluorescein isothiocyanate (FITC)-dextran was examined in the ONL, indicating the outer BRB breakdown and RPE dysfunction. The breakdown of outer BRB was due to the decreased protein levels of zonula occludens-1 (ZO-1) and occludin in the RPE-Bruch’s membrane-choriocapillaris complex, which was caused by the activated hypoxia inducible factor 1α (HIF-1α) and c-Jun N-terminal kinase (JNK) pathways [39]. One previous study reported that high-glucose induced upregulation of claudin-1, which decreased apical-basolateral permeability in ARPE-19 cell line; however, the transepithelial permeability of ARPE-19 cells was increased in high glucose treated ARPE-19 cells [52]. In hyperglycemia cultured bovine RPE cells, Na^+^-K^+^-ATPase function was impaired, which could be restored by the aldose reductase inhibitor [53]. Therefore, it appears that hyperglycemia impairs the transport of water from subretinal space to the choriocapillaris, contributing to DME development. RPE degeneration, as evidence with electron microscopy in diabetic animal models demonstrating shrunken nuclei, reduced endoplasmic reticulum, infolding of cell membranes, altered melanosomes and reduced number of RPE cells [54], further aggregating the breakdown of outer BRB. OCT characteristics of the outer retina in DME have proven that ellipsoid zone (EZ) disruption occurs subsequent to the disruption of the external limiting membrane (ELM) [55]. Since the ELM, the apical microvilli of Müller cells, functions as the third retinal barrier, ELM disruption may result in photoreceptor and RPE damage. In fact, decreased RPE thickness was observed in DME patients with non-proliferative DR (NPDR) or proliferative DR (PDR) [56]. Thus, both inner and outer BRB breakdown in DR contributes the fluid accumulation in macular region, leading to DME formation.

### 3.4. Inflammatory Effect Contributing to DME Formation

DR is considered as a disease of chronic microinflammation [57,58]. Accumulating evidence showed that immunological and inflammatory mechanisms play a prominent role in the pathogenesis of DR and DME [2,18]. In DR, the microinflammation, a low-grade background inflammation, is maintained by cytokines such as IL-6, IL-8 and monocyte chemoattractant protein-1 (MCP-1, also known as the C-C motif chemokine ligand 2 (CCL2)) [18,59]. IL-6 alters the function of the astrocytes, leading to disruption of the inner BRB. IL-8 and MCP-1 act on neutrophils and monocytes, promoting infiltration of these cells into the retina. Alongside inflammation-related factors, inflammatory cells, such as leukocytes and microglia, also play pivotal roles in DME.

#### 3.4.1. Inflammation-Related Factors Are Upregulated in DME

Leukocyte adhesion and the subsequent leukostasis is the early inflammatory response in DME. In DR, endothelial cells upregulated ICAM-1 expression, leading to increased leukocyte adhesion, resulting in retinal vascular leakage [60]. In diabetic animals, neutrophils increased the expression of integrin such as CD18, and exhibited higher integrin-mediated adhesion, whereas, antibodies against CD18 or ICAM-1, and knockout of these genes inhibited the leukostasis and decreased BRB breakdown [61]. In DR, chronic hyperglycemia increases expression of chemokines, including MCP-1/CCL2, that increase and mediate leukocyte adhesion, leukostasis, and infiltration of monocytes into the retinal parenchyma [16]. The infiltrated monocytes differentiate into activated macrophages and produce many cytokines and inflammation-related factors, including VEGF, IL-6, TNF-α and angiopoietin-2 (Ang-2) [41]. These cytokines and mediators disrupt the cell-cell junctions, resulting in BRB breakdown. BRB breakdown further aggregates the over-production of the inflammation-related factors. The inflammation-related factors contributing to BRB breakdown include but are not limited to VEGF, TNF-α, IL-1α, IL-1β, IL-6, IL-8, IL-10, ICAM-1, MCP-1, placental growth factor (PlGF), hepatocyte growth factor (HGF), insulin-like growth factor-1 (IGF-1), histamine and complement factors, which were reported to be upregulated in the vitreous or aqueous humor of DME patients [18,60], indicating the significant contribution of inflammation-related factors to the pathogenesis of DME.

#### 3.4.2. Inflammatory Cells Are Activated in DME

In addition to inflammation-related factors, inflammatory cells, such as activated microglia, also contribute to the pathogenesis of DME and DR [62]. In diabetic eyes, hyperglycemia-induced activation of retinal microglia and infiltration of immune cells, including macrophages, lymphocytes, and neutrophils, have been reported [63,64,65].

In the retina, resident microglia, regarded as the immunological guard, are sensitive to the changes in the retinal microenvironment and respond quickly to various insults [66,67]. Microglia are activated early in streptozotocin-induced diabetic rat retinas [68]. Activated microglia, with a more amoeboid phenotype with increased motility, migrate from inner retina to the outer retina in experimental DR. The activated microglia, releasing the inflammatory factors and phagocytosing the apoptotic neurons, contribute to the anatomical and functional abnormalities in diabetic retina [69]. The phenomena of microglia activation in diabetic retina has been documented by our recent work [70,71]. In the diabetic rat, microglia became activated with increased cell proliferation, close contact with the retinal capillaries, especially the deep capillary plexus, and enhanced migration from inner to outer retina, even to the subretinal space and RPE layer [70]. Our observation showed that the activated microglia penetrated the basement membrane of the capillaries and phagocytosed endothelial cells, leading to BRB breakdown and the formation of acellular capillaries [70]. This enhanced phagocytosis by microglia was associated with decreased Src/Akt/Cofilin pathway signaling [70]. 

Alongside direct interaction, activated microglia mediate the death of retinal ganglion cells (RGCs) non-cell-autonomously by releasing TNF-α [72]. In experimental DR, activated microglia synthesize inflammatory factors, cytokines, proteases, nitrous oxide, and ROS, accompanying with neuronal death in DR [66,69]. In a proof-of-concept finding, the increased inflammatory factors and intracellular ROS could be prevented in experimental DR by intravitreal injection of fractalkine (FKN), an inhibitor of microglia activation [71]. Graeber et al. observed the number of holes in the RPE layer was increased in diabetic mice model [73], which facilitated the migration of the inflammatory cells. Activated microglia recruit macrophages and other inflammatory cells into the retina by increasing the release of CCL2/MCP-1 [16,74].

In patients with DR, activated microglia, increased in number and becoming hypertrophy, clustered around the retinal vessels, microaneurysms, intraretinal hemorrhages, and cotton-wool spots, etc. [75]. Activated microglia even migrate to the outer retina and subretinal space in some retinas with cystoid macular edema [75]. Recently, hyperreflective foci (HRF) viewed with OCT or OCTA in the diabetic retina have been considered as a biomarker of active inflammatory cells, especially microglia and/or macrophages. One previous study demonstrated a positive correlation between the level of aqueous CD14, the cytokine released by microglia and macrophages, and the increased HRF number in DME patients, indicating that inflammatory cells, such as microglia, participate in the pathogenesis of DME [62]. 

The above findings indicate that DR is an inflammatory disease, characterized by increased proinflammatory mediators and activated cellular inflammatory processes resulting in BRB breakdown and DME formation [60,61,76,77,78,79]. 

### 3.5. Diabetic Retinal Neurodegeneration (DRN) Aggregating the Functional Outcome in DME

DRN involves degenerative alterations in retinal neurons and glia, including RGCs, photoreceptors, amacrine cells, bipolar cells and glial cells. The characteristics of DRN include increasing neural cell apoptosis, reduced retinal function, and reactive gliosis. DRN might be driven by accumulating glutamate, inflammation, oxidative stress, and altered balance of neurotrophic factors in retina. It should be noted that DRN and diabetic retinal microangiopathy are distinct but interdependent components of DR [80,81,82,83]. 

Neurodegeneration, characterized by an increase in neuronal apoptosis, could be detected in the early stage of DR [84]. In diabetic retina, a high apoptotic rate of RGCs was reported [84,85]. In diabetic donors, RGCs showed cytoplasmic immunoreactivity for caspase-3, Fas, and Bax [86]. In ONL, increasing apoptosis of photoreceptors was also observed between 4 and 24 weeks after diabetes onset in diabetic rats [87,88]. In the eyes of patients with NPDR, the immunohistochemistry showed the association of upregulated expressions of Bax, caspase-9 and -3 with neuronal cell death [89].

In a prospective longitudinal study including 45 T1DM patients with no or minimal DR, OCT showed the thickness of NFL, ganglion cell layer (GCL), and inner plexiform layer (IPL) decreased progressively over time [90]. In this study, after adjustment for diverse factors, an obvious, progressive loss of NFL (0.25 μm/year) both parafoveally and perifoveally and loss of the GCL+IPL (0.29 μm/y) parafoveally was detected in patients with T1DM [90]. In a separate observation comparing 5 donor eyes with diabetes mellitus and no or minimal DR (diabetic group) to 5 age-matched donor eyes without diabetes mellitus (control group), the thickness of NFL was obviously decreased (17.3 μm) in diabetic group than that in control group (30.4 μm), while the retinal vascular density remained relatively unchanged between the two groups [90]. The above observations suggested that diabetes could directly affect the neuroretina in addition to retinal capillaries. The neuroprotection should not be ignored during the management of DME.

### 3.6. Proteomics and Metabolomics Leading to Deep Understanding and Targeted Treatments for DME

In DR and DME, chronic hyperglycemia alters retinal homeostatic mechanisms, resulting in the changes in the proteomic and metabolomic microenvironment, which is crucial to retinal cell function. With the rapid advancement in methodology, numerous studies on proteomics and metabolomics have been conducted on samples from DR and DME patients and shed a light in the pathogenesis of DR and DME, advancing and expanding our understanding of the pathogenesis of DR and DME. 

The changes in proteomics in DR and DME have expanded our recognition of this blinding disease using proteomic approaches, e.g., two-dimensional difference gel electrophoresis (2D-DIGE) coupled with mass spectrometry (MS), sodium dodecyl sulfate-polyacrylamide gel electrophoresis (SDS-PAGE) coupled with MS [91], liquid chromatography coupled with tandem MS (LC-MS/MS) [92], and bead-based multiplex immunoassays [93], and etc. With the advancing methods detecting the proteomics in vitreous humor of DR patients, several proteins, such as complement component C3, ICAM-1, IL-6, serum amyloid A protein (SAA), amyloid-β A4 protein, kininogen-1, metalloproteinase inhibitor 1, and VEGF, etc., have been identified as potential biomarkers for different stages of DR [94]. Thus, a thorough understanding of the proteomic changes could provide new insight into elucidation of the pathogenesis as well as biomarker for DR and DME, which would lead to the development of potential treatments for DR and DME. 

Using the proteomics technologies, the potential biomarker for DR could be discovered. The vitreous samples from pre-proliferative DR associated with DME (DME group) and without DME (non-DME group) were analyzed for the changes in proteomics. The results showed that a total of 14 proteins (DME group) and 15 proteins (non-DME group) were differentially expressed. Further analysis showed six proteins were upregulated, i.e., pigment epithelium-derived factor (PEDF), apolipoprotein A (ApoA)-1 (ApoA-1), ApoA-4, thyroid hormone receptor- interacting protein-11 (Trip-11), plasma retinol-binding protein (PRBP), and vitamin D binding protein (VDBP) in DME group; while apolipoprotein H (Apo H) was expressed only in non-DME group [95]. The differential changes in proteomics indicate the biomarker for DME group and also demonstrated the involvement of these molecules in the pathogenesis of DME. A recent study using LC-MS/MS analysis to detect the proteomics in aqueous humor in 73 eyes of DME patients with NPDR or PDR showed that about 308 significantly changed proteins between NPDR group and PDR group [96]. Compared with NPDR group, in PDR group, the upregulated proteins are involved in the immune system and/or negative regulation of the cell cycle, while the downregulated proteins are related with the VEGF receptor (VEGFR) pathway and/or metabolism [96]. Further analysis showed that, compared with the NPDR and non-diabetic groups, the immune-associated protein cystatin C (CST3), downregulated in the PDR group, might be served as a novel target in DME treatment [96]. 

Metabolomics, the qualitative and quantitative assessment of the metabolites in body fluids [97], is a promising branch of omics to detect the changes in metabolites and the underlying mechanisms in various diseases, including DR and DME [98]. Metabolomics could provide the novel metabolic biomarkers and potential pathways in DR and DME, facilitating the elucidation of the mechanisms, and proposing new therapeutic strategies for DR and DME. One current review reported that several metabolites, such as L-glutamine, L-lactic acid, pyruvic acid, acetic acid, L-glutamic acid, D-glucose, L-alanine, L-threonine, citrulline, L-lysine, and succinic acid, were potential biomarkers of DR [99], which involved new pathogenic pathways in DR. Alongside the ocular samples, the metabolites in serum and metabolic pathways were detected and compared between different stages of DR in patients with T2DM [100]. The results showed that the pathways including arginine biosynthesis metabolism, linoleic acid metabolism, glutamate metabolism, D-glutamine and D-glutamate metabolism, etc., were dysregulated in DR patients of the Asian population [100]. The metabolic signatures, such as upregulated levels of glutamate, aspartate, glutamine, N-acetyl-L-glutamate, and N-acetyl-L-aspartate, and downregulated levels of dihomo-gamma-linolenate, docosahexaenoic, and eicosatetraenoic, can be employed to differentiate PDR from NPDR in the Asian population [100]. 

Thus, with the advancements in biotechnologies of proteomics and metabolomics, the key new pathways, biomarkers, and establishing therapeutic targets would be in the practice in the management of DR and DME, further enhancing our understanding of this disease. 

## 4. Therapeutic Strategies for DME

Since the pathogenesis of DME involves different mechanisms with multiple factors and pathways participation, the treatment of DME should be the multimodality therapies, comprising the systemic control of the risk factors, regulating the potential targets, anti-inflammation, anti-oxidative stress, neuroprotection, laser and subthreshold micropulse laser therapy, photobiomodulation, as well as the vitrectomy, etc. (Figure 1).

### 4.1. Control of Systemic Risk Factors

Since DME remains a common complication of DR caused by diabetes, control of systemic risk factors including tight control of hyperglycemia, hyperlipidemia and hypertension should be considered as the fundamental strategy for the prevention and treatment of DR and DME. The Diabetes Control and Complications Trial (DCCT) showed that intensive glycemic control in T1DM reduced the risk of developing retinopathy by 76% and also reduced the risk of proliferative disease and the need for laser treatment [101]. For patients with T2DM, the UK Prospective Diabetes Study (UKPDS) showed that intensive glycemic control can reduce the need of laser photocoagulation treatment and decrease the risk of progression to blindness [102]. The UKPDS analyzed the effect of intensive control of blood pressure with captopril or atenolol on microvascular complications in 1148 hypertensive patients with T2DM [103]. The results showed that tight control of blood pressure decreased the risk of complications related to diabetes, progression of DR, and deterioration in visual acuity. The Action to Control Cardiovascular Risk in Diabetes (ACCORD) Eye Study showed a 40% reduction in relative risk of retinopathy progression with the addition of fenofibrate to simvastatin to control blood lipids [104]. Thus, the intensive control of hyperglycemia, hypertension and hyperlipidemia seems to reduce the rate of progression of DR, which will also benefit the treatment outcome of DME. Furthermore, erythrocyte membrane fluidity, might be served as a novel biomarker to predict the risk of developing complications in T1DM patients supplementing hemoglobin A_1c_ in long-term T1DM management [44].

### 4.2. Laser Therapy

The efficacy and safety of focal laser for treating DME was validated by the Early Treatment of Diabetic Retinopathy Study (ETDRS) in the 1980s [105]. Today, the focal/grid laser is an alternative in eyes with DME, mostly for non-center involved DME (Non-CI-DME). The subthreshold micropulse laser has been accepted as a potential and promising treatment in some cases for DME [106], due to its safe, non-scarring alternative procedure [107,108,109,110]. Subthreshold micropulse laser therapy is known to improve RPE function, modulate the activation of heat-shock proteins and normalize cytokine expression [111], and it seems to result in the normalization of retinal neuroinflammatory metabolic pathways [112]. Both the infrared subthreshold micropulse laser (810-nm wavelength) and yellow subthreshold micropulse laser (577-nm wavelength) were shown to be effective for DME treatment with good safety [113,114]. In a prospective study, the morphological changes in the retina and choroid, and the function of macula were evaluated and compared in patients with center-involved DME (CI-DME), who were treated with yellow (577-nm) or infrared (810-nm) subthreshold micropulse laser [115]. The data showed that both treatments are safe based on the morphological and functional evaluations in mild CI-DME [115]. 

### 4.3. Intravitreal Injection of Anti-VEGF Agents

Since many cytokines and various pathways are implicated in the pathogenic process of DME, treating targets become the fundamental strategy for DME treatment [116]. In the pathogenesis of DR and DME, VEGF is upregulated and plays a pivotal role leading to BRB breakdown, macular edema, and neovascularization [117], and the severity of leakage in DME correlates with the level of VEGF [118]. Targeting VEGF (anti-VEGF) treatment has demonstrated significant benefits for patients with DME, which has become the first-line treatment for DME [116], supplanting focal photocoagulation. Currently, there are several anti-VEGF agents which are commercially available for DME, including Lucentis (ranibizumab), Eylea (aflibercept), Lumitin (conbercept), Beovu (brolucizumab), and off-label Avastin (bevacizumab) [119,120,121,122,123,124], differing in molecular weight and structure, binding affinity, targeted VEGF isoforms, and concentration, etc. For instance, brolucizumab treatment improved visual function and retinal morphology significantly in DME patients with a favorable benefit/risk profile [125]. Anti-VEGF drugs maintain BRB integrity by antagonizing VEGF-A and/or PlGF. However, some patients respond unsatisfactorily to anti-VEGF therapy. There are 31.6~65.6% patients with DME who still exist with persistent edema even after regular intravitreal injection for at least 4 injections within 24 weeks [19]. Furthermore, many DME patients need repeated injections with high cost due to multiple injections, and some patients even respond incompletely or are unresponsive to anti-VEGF treatment (non-responders). These limitations and suboptimal responses to anti-VEGF treatment indicate other pathogenic factors, beyond VEGF, might contribute to the pathogenesis of DME, which prompted the researchers to develop novel therapeutic approaches. 

### 4.4. Emerging Therapeutic Strategies Targeting VEGF/VEGFR System and the Accessory Proteins

The current trend for anti-VEGF development is toward either smaller molecular weight targeting VEGF-A (e.g., Beovu and abicipar), fusion proteins targeting VEGF-A in combination with other factors (e.g., faricimab), targeting other VEGF family members (OPT-302), reducing the cost of burden (developing biosimilars), or improving treatment durability (KSI-301, port delivery system, gene therapy), and etc.

#### 4.4.1. Abicipar Pegol

Abicipar pegol (AGN-150998, Allergan plc/Molecular Partners) belongs to a family of the designed ankyrin repeat proteins (DARPin). Abicipar pegol binds VEGF-A with high affinity [126]. Compared with ranibizumab, abicipar pegol improved its pharmacokinetic properties, i.e., lower molecular weight (34 vs. 48 kDa), higher target binding affinity (2 vs. 46 pM) and longer ocular half-life (≥13 vs. 7 days in the aqueous humor) [127,128,129]. In phase I/II, open-label, multicenter dose-escalation trial for DME, prolonged edema reduction and visual improvement was achieved in several patients, however, ocular inflammation was a major concern [128]. Phase III clinical trials, SEQUOIA (NCT02462486) and CEDAR (NCT02462928), showed non-inferior result in visual acuity improvement compared with monthly ranibizumab injection in patients with neovascular age-related macular degeneration (nAMD) [130,131]. However, it failed to gain FDA approval due to significant intraocular inflammation and an unfavorable risk-benefit ratio [130,132]. Further studies are needed for evaluation of the efficacy and safety of abicipar pegol in DR treatment before its approval [133]. 

#### 4.4.2. OPT-302

OPT-302 (Opthea; Victoria, Australia) is a soluble form of VEGF receptor 3 (VEGFR-3) consisting of the extracellular domains 1–3 of human VEGFR-3 and the Fc fragment of human IgG1. OPT-302 blocks the activity of the proteins VEGF-C and VEGF-D [134], which may serve a complementary therapeutic role in VEGF-mediated DR pathogenesis, and overcome the limitation of the current anti-VEGF drugs that only target VEGF-A. Intravitreal OPT-302 was safe and well tolerated, and the combined treatment with OPT-302 may enhance the efficacy in neovascular suppression in nAMD [134]. A multicenter phase 1b/2a trial has evaluated OPT-302 in combination with aflibercept for refractory DME [135]. Combo-therapy using OPT-302 and aflibercept or conbercept may target all the VEGF family members, which might be effective in patients with retinal vascular diseases and is worth trying. 

#### 4.4.3. Anti-VEGF Biosimilars

The anti-VEGF medications have been available for more than a decade and their patent expiration dates are coming. For example, ranibizumab’s patent expired in June 2020 in the United States (2022 in the European Union) and aflibercept’s patent will expire in 2023 in the United States (2025 in the European Union) [136]. With the expiry of these patents, the transition to biosimilars can have a significant impact worldwide due to the favorable cost-effectiveness [137]. 

Many bevacizumab biosimilars were approved for cancer treatment. However, due to a cheaper alternative to ranibizumab, the off-label use of bevacizumab is still increasing in ophthalmology [137]. Currently, there are several anti-VEGF biosimilars to ranibizumab and aflibercept in the development stage or acquiring approval [137]. For example, Razumab^®^ (Intas Pharmaceutical Ltd., Ahmedabad, GJ, India) is the first biosimilar to ranibizumab approved for ophthalmic use in India by the drug controller general of India for nAMD, myopic choroidal neovascularization, DME, and retinal vein occlusion-macular edema (RVO-ME) [137,138,139,140]. In 2021, the US FDA approved Byooviz (ranibizumab-nuna) as the first biosimilar to Lucentis (ranibizumab injection) for the treatment of several eye diseases, including nAMD, RVO-ME, and CNV. As for aflibercept biosimilars, ABP-938 (Amgen, Thousand Oaks, CA, USA) is under phase 3 trial, which is scheduled to be completed by July 2023 (NCT04270747). Other biosimilars to aflibercept (MYL-1710P, ABP-938, and CHS-2020, USA; FYB203, Germany; SB15, South Korea) underwent the clinical trials [141]. 

#### 4.4.4. KSI-301

KSI-301 (KODIAK sciences, Palo Alto, CA, USA) comprises a specific anti-VEGF IgG1 antibody and an inert immune effector, covalently linked to a high molecular weight phosphorycholine biopolymer (950 kDa). Intravitreal injection of KSI-301 showed prolonged intravitreal half-life (about 6 months) due to slow diffusion and decreased elimination in the eye [142,143]. Clinical trials (GLEAM Study and GLIMMER study) are underway. The patients are randomized into two groups receiving either intravitreal KSI-301 or aflibercept [144]. Phase 2b/3 clinical trial failed to meet the primary endpoint of visual acuity gains in nAMD patients treated with KSI-301 compared to aflibercept [145]. 

#### 4.4.5. Port Delivery System (PDS) with Ranibizumab

Currently, the delivery of anti-VEGF drugs is largely dependent on repeated intravitreal injections. PDS allows continuous release of ranibizumab, and minimizes the need for frequent injections [146]. Sustained and controlled release is achieved by the porous metal element allowing passive diffusion of drugs from PDS to the vitreous [147]. ARCHWAY (NCT03677934) randomized Phase 3 trial of PDS with ranibizumab showed that PDS with ranibizumab met its primary objective, demonstrating equivalent efficacy of monthly ranibizumab injection [148]. Phase 3 clinical trials for DR (PAVILION; NCT04503551), and DME (PAGODA; NCT04108156) are currently in progress. 

#### 4.4.6. High-Dose of Anti-VEGF Agents

An intravitreal injection of high-dose anti-VEGF agents might prolong the intravitreal injection intervals and improve drug efficacy. Using rabbits, Kim et al. showed that a two-fold increase in retinal half-life and prolonged effective concentration of ranibizumab in retina when administered a 10-fold dose of ranibizumab with good safety in rabbit eyes [149]. Currently, phase 3 clinical trials are underway in DME (PHOTON; NCT04429503) and nAMD (PULSAR; NCT04423718). 

#### 4.4.7. Gene Therapy to Deliver Anti-VEGF Agents

Given the burden of repeated anti-VEGF treatments, gene therapy can achieve long-term expression of anti-VEGF proteins to suppress of VEGF in retinal vascular diseases. Several gene therapy drugs, including RGX-314, ADVM-022 and rAAV-sFlt1, are currently under clinical evaluation. 

RGX-314 is an adeno-associated virus 8 (AAV8) vector encoding ranibizumab. The Phase II ALTITUDE trial is studying the patients with DR but without DME, who are treated with a single dose of RGX-314, delivered in suprachoroidal space [150]. Positive 3-month interim data from cohort 1, treated with a single injection at a dose of 2.5 × 10^11^ genomic copies per eye, showed that treatment was well-tolerated and 33% of patients had a ≥ 2-step improvement from baseline [150].

ADVM-022, an AAV2-7m8 vector encoding aflibercept, is optimized for intravitreal delivery. Prolonged expression and efficacy of ADVM-022 was evaluated in a laser-induced CNV model in non-human primates with promising outcomes [151]. Clinical trials for nAMD (NCT04645212; NCT03748784) and DME (NCT04418427) are currently underway, evaluating safety and efficacy following a single intravitreal injection of ADVM-022.

rAAV-sFlt1, a recombinant AAV2 vector expressing soluble VEGF receptor 1, works as a decoy receptor for VEGF. A pre-clinical study showed safety and well-toleration in non-human primates after a single subretinal injection of rAAV-sFlt1 [152]. Although phase I study (NCT01494805) demonstrated the safety in nAMD patients [153], phase IIa clinical trial (NCT01494805) showed no obvious benefit in visual acuity or anatomy [154]. The potential effect of rAAV-sFlt1 on DME deserves further study [155].

#### 4.4.8. Targeting VEGFRs

The inhibition of VEGFRs is one of the promising strategies for treatment of VEGF-driven neovascular diseases [156,157,158,159,160]. Targeting VEGFRs has been extensively studied in clinical oncology. There are several approaches to inhibiting VEGFR signaling, i.e., VEGFR antibodies, VEGFR allosteric inhibitors, and inhibition of the intracellular tyrosine kinase of VEGFR by tyrosine kinase inhibitors (TKIs).

Ramucirumab (Cyramza^®^), a fully humanized anti-VEGFR-2 monoclonal antibody, was approved for the treatment of cancer patients who experience disease progression during chemotherapy [158]. Its ophthalmic use is to treat retinal vascular diseases, including DME, which deserves further exploration. 

GB-102 (GrayBug Vision; Redwood City, CA, USA), sunitinib maleate and a TKI with activity against both VEGF-A and PDGF, is encapsulated within bioerodible polymer nanoparticles degrading slowly over time [161]. Single GB-102 treatment can last up to 6 months with comparable visual acuity and CSFT outcomes [162,163]. Phase 1/2a study (ADAGIO) reported that the majority of nAMD patients were maintained at 3 months (88%) and 6 months (68%) with a single dose of GB-102 [164]. Among the emerging therapies, GB-102, with longer duration between treatments, would impact significantly on the patient’s life with less frequent follow-up and less expenditure [164].

X-82 (Tyrogenex) is an oral anti-PDGF and VEGF-A inhibitor. In a Phase 1 study (NCT02348359) for nAMD, 29% patients (10 of 35) did not complete the 24-week endpoint, with 6 (17%) withdrawing due to adverse events, including diarrhea, nausea, fatigue, and transaminase elevation [165]. Phase 2 APEX study (NCT02348359) is underway, which compares X-82 (Tyrogenex) with as-needed aflibercept injections to aflibercept monotherapy.

PAN-90806 (PanOptica; Mount Arlington, NJ, USA), a TKI eyedrop, was shown to inhibit VEGF signaling with topical once daily dosing. According to a phase 1/2 study (NCT03479372), PAN-90806 showed favorable safety and effectiveness as monotherapy. However, it may be applicable for certain patients and further studies are needed [162].

#### 4.4.9. Targeting Neuropilin-1

Vesencumab is a human IgG1 monoclonal antibody against neuropilin-1 (NRP-1), with potential anti-angiogenic and anti-neoplastic activities. Vesencumab specifically targets and binds to NRP-1, preventing the subsequent coupling of NRP-1 to VEGFR-2, thereby decreasing VEGF-mediated signaling. When combined with other anti-VEGF therapies, vesencumab may enhance their anti-angiogenic effect [166]. Vesencumab is currently undergoing clinical study for cancer patients [167].

### 4.5. Anti-Inflammatory Therapy

Since inflammation plays a critical role in DR and DME, suppression of inflammation seems to be a reasonable approach for treating DR and DME [57]. Corticosteroids have been proven to be beneficial in treating DR and DME due to their anti-inflammatory and anti-angiogenic properties [168]. At present, intravitreal preservative-free triamcinolone, the extended-release dexamethasone implant (Ozurdex) and the fluocinolone acetonide implant (Iluvein) are FDA-approved for treating DME. Intravitreal injection of sustainable dexamethasone (Ozurdex, Allergen) was safe and effective in DME treatment, achieving visual improvement, reducing edema, and decreasing the inflammatory cytokines, such as VEGF, MCP-1, and IL-6. A MEAD study (NCT00168337 and NCT00168389) evaluated the safety and efficacy of Ozurdex (0.7 mg and 0.35 mg) and demonstrated both doses of the Ozurdex implant met the primary objective for visual improvement with acceptable safety profile [169]. A FAME study evaluated long-term effects, including efficacy and safety, of intravitreal inserts of fluocinolone acetonide (Iluvien, Alimera Sciences, Alpharetta, GA, USA), releasing 0.2 μg/d (low dose) or 0.5 μg/d (high dose) for DME treatment. Iluvein inserts provided substantial visual benefit for up to 3 years [170], especially for those who are unresponsive to other therapy, such as anti-VEGF treatment [171]. However, due to concerns regarding the intraocular pressure elevation and cataract formation, corticosteroids are used as the second-line therapy for DME.

Based on the inflammatory theory of DME formation, the ongoing translational research targeting inflammatory cells and factors is shedding new light on the management of DME beyond anti-VEGF therapy. Anti-inflammation treatment can be roughly classified into several categories, i.e., regulation/inhibition of inflammatory cells (such as minocycline, dextromethorphan), targeting various inflammatory mediators (such as corticosteroids, TAK-779, TNF-α inhibitor, IL-6/IL-6R inhibitor, non-steroid anti-inflammatory drugs (NSAIDs)), and alternative delivery route (Suprachoroidal injection, oral and subcutaneous injection), and etc. 

#### 4.5.1. Minocycline and Dextromethorphan

Minocycline, besides its antimicrobial activity, has anti-inflammatory, anti-oxidant, anti-apoptotic, neuroprotective, and immunomodulatory effects [172]. In phase I/II clinical trial (ClinicalTrials.gov number, NCT01120899), oral minocycline treatment improved visual function and reduced central macular edema [173]. Dextromethorphan was effective in decreasing vascular leakage in 5 DME patients in phase I/II clinical trial, in which oral dextromethorphan was administered 60 mg twice daily for 6 months as monotherapy [174]. 

#### 4.5.2. Difluprednate and Dexamethasone-Cyclodextrin

Difluprednate (difluprednisolone butyrate acetate, DFBA) is an anti-inflammatory steroid, effective in the treatment of anterior uveitis, postoperative ocular inflammation, and pain [175,176]. Difluprednate ophthalmic emulsion 0.05% (Durezol (TM), Sirion Therapeutics Inc., Tampa, FL, USA) effectively reduces refractory DME post-vitrectomy [177], and diffuse DME without surgical intervention [178]. Topical dexamethasone-cyclodextrin eye drops were safe, improved visual acuity and decreased central macular thickness in DME patients [179]. In a randomized, controlled trial, topical dexamethasone-cyclodextrin nanoparticle eye drops (1.5%) significantly improved the vision and decreased macular thickness in DME patients [180]. 

#### 4.5.3. TAK-779

TAK-779, a dual CCR2/CCR5 inhibitor, significantly reduced retinal vascular permeability in diabetic mice [181]. TAK-779 also decreased infiltration of macrophage/microglia, reduced the expressions of ICAM-1 and stromal cell-derived factor 1 (SDF-1), and restored zonula occludens-1 (ZO-1) in diabetic mouse retina [181]. Targeting CCR2/CCR5 might provide a novel strategy for DME management. 

#### 4.5.4. Targeting Integrin

Integrins are involved in many biological processes and play a critical role in the pathogenesis of many diseases. Some integrins are associated with vitreolysis, angiogenesis, and ocular surface diseases [182]. Anti-β2-integrin or anti-ICAM-1 decreased leukocyte adhesion, the death of endothelial cells, and BRB breakdown [183,184,185]. Therefore, targeting integrins, independent of anti-VEGF therapies, has the potential to prevent vision loss. 

Risuteganib (Luminate, ALG-1001, Allegro Ophthalmics, LLC, San Juan Capistrano, CA, USA) is an engineered arginyl-glycyl-aspartic acid (RGD) class synthetic peptide targeting integrin. RGD peptide treatment suppressed retinal neovascularization and released cellular adhesion to induce posterior vitreous detachment [186,187]. Risuteganib has potential as a therapy for DR and DME [182,188]. SB-267268 (GlaxoSmithKline) is a small molecule inhibitor of αvβ3 and αvβ5 integrins [189]. In an animal model of retinopathy of prematurity (ROP), SB-267268 decreased the mRNA expressions of VEGF and VEGFR2, and reduced the pathological angiogenesis by 50% [189]. 

#### 4.5.5. Targeting TNF-α

TNF-α is an inflammatory cytokine that promotes the upregulation of adhesion molecule expression, leukocyte recruitment and monocyte attraction. TNF-α was increased in the aqueous and vitreous of diabetic patients compared to control subjects [190,191,192]. The targeting TNF-α might provide an option for treating DR and DME. Currently, there are monoclonal anti-TNF-α full IgG1 antibodies (infliximab, adalimumab, and golimumab), PEGylated Fab’ fragment of anti-TNF-α antibody (certolizumab pegol) and extracellular domain of TNF receptor 2/IgG1-Fc fusion protein (etanercept), effective for the treatment of rheumatoid arthritis [193]. In fact, a clinical study with infliximab achieved functional and anatomical improvement in DME patients, highlighting the pathogenic role of TNF-α in DR [194].

#### 4.5.6. Targeting IL-6/IL-6R

IL-6/IL-6R exerts an important role in initiating the breakdown of BRB in DR [195,196], due to the disrupting of the barrier function and increasing vascular leakage via the downregulation of tight junction proteins [197]. IL-6 signaling occurs through its membrane-bound receptor IL-6R (classical signaling) or through the soluble IL-6R (sIL-6R, trans-signaling) [198,199]. Anti-IL-6 and anti-IL-6R strategies target both classical and trans-signaling pathways to block IL-6 signaling. Several therapeutic strategies targeting IL-6 signaling pathways are in progress [200], including anti-IL-6 antibodies (e.g., siltuximab and sirukumab), anti-IL-6R antibodies (e.g., tocilizumab and vobarilizumab), and IL-6 trans-signaling selective inhibitor (olamkicept). Tocilizumab is effective in the treatment of various autoimmune and inflammatory diseases, including rheumatoid arthritis, with a favorable outcome [201]. Thus, blocking IL-6 and IL-6R may be potential approaches for treating DR.

#### 4.5.7. Vascular Adhesion Protein-1 (VAP-1) Inhibitor

VAP-1, known as amine oxidase copper-containing 3 (AOC3) and semicarbazide-sensitive amine oxidase, is a membrane-bound adhesion protein facilitating leukocyte adhesion and transmigration to the inflammatory site [202]. Previous studies showed that the level of soluble VAP-1 was higher in the vitreous of PDR patients than in that of nondiabetic patients [203]. In diabetic rats, the leukocyte transmigration rate was reduced by UV-002 (a specific inhibitor of VAP-1) [202]. In diabetic animals, VAP-1 inhibition improved retinal function and structure as evidenced by electroretinogram and histopathological studies [204]. Thus, VAP-1 could be an underlying target for DR treatment [205]. A phase 2 study (VIDI study, NCT02302079) tested the effect of ASP8232, a specific VAP-1 inhibitor, on CI-DME [206]. The primary data showed that ASP8232 nearly inhibited the activity of plasma VAP-1, while had no effect on CSFT in patients with CI-DME. The clinical application of VAP-1 inhibition still requires further study.

#### 4.5.8. Non-Steroid Anti-Inflammatory Drugs (NSAIDs)

NSAIDs inhibit the cyclooxygenase (COX) enzyme that is an essential mediator through the regulation of prostaglandin dependent pathways [207]. Bromfenac mainly inhibits the activity of COX-2 [208], and nepafenac, a prodrug, acts on COX-1 and COX-2 through its active metabolite amfenac [209]. NSAIDs were reported effective in DME with various and heterogeneous results. In a pilot study, topical bromfenac significantly reduced central macular thickness in patients with DME, however, without obvious effect on visual acuity [210]. The safety and efficacy of topical nepafenac 0.1% were tested in 6 eyes of 5 patients with DME, which showed that topical nepafenac treatment improved vision and decreased retinal thickness [211]. Postoperative topical nepafenac was shown to be effective for prophylaxis of macular edema in diabetic patients underwent phacoemulsification and intraocular lens implantation [212]. Further investigations on whether topical NSAIDs could serve as an alternative or adjunctive treatment to intravitreal anti-VEGF therapy are required.

#### 4.5.9. Suprachoroidal Injection of Steroid

The suprachoroidal space (SCS) has become an applicable route to deliver drugs to the back of the eye via suprachoroidal injection [213]. When delivered to the suprachoroidal space, the drug can target both the retina and the choroid, overcoming multiple ocular tissue barriers and achieving the efficacy at low dose [142,214]. The phase 2 TYBEE clinical trial enrolled 71 eyes with treatment-naïve DME [215], with 36 eyes received SCS injection of triamcinolone acetonide (TA) (CLS-TA, 4 mg/100 µL) and aflibercept (2 mg/0.05 mL) at baseline and week 12 (active group) and 35 eyes which were treated with aflibercept (control group). At 24 weeks from baseline, the visual acuity gain was similar between two groups, with mild anatomic improvement and less treatment burden in the active group [215]. 

### 4.6. Targeting Ang-2/Tyrosine Kinase with Immunoglobulin-like and Epidermal Growth Factor-like Domains 2 (Tie2) System

The angiopoietin (Ang)/Tie2 pathway is involved in many retinal vascular diseases. Angiopoietin-1 (Ang-1) and Ang-2 ligands compete for the Tie2 receptor. Tie2 is a tyrosine kinase receptor in vascular endothelial cells and maintains vascular stability. Tie2 activation by Ang-1 increases the survival, adhesion, and cell junction integrity of endothelial cells, while Ang-2 interferes with the Ang-1/Tie2 axis, resulting in vascular instability. Vascular endothelial-protein tyrosine phosphatase (VE-PTP) is an endothelial cell-specific phosphatase, which forms a complex with Tie2 and dephosphorylates Tie2, against the actions of Ang-1 [216]. In DR, there is an increased production of Ang-2, competitively binding Tie2 to reduce Ties phosphorylation, whereas VE-PTP directly decreases Tie2 phosphorylation. Tie2 inactivation destabilizes the vasculature, resulting in pericyte dropout, reduction in endothelial cell viability, decreased endothelial cell anchor and cell junction integrity. Thus, activating the Tie2 signaling pathway, by the inhibition of Ang-2 or VE-PTP, should be a therapeutic strategy for retinal vascular diseases.

#### 4.6.1. Targeting Ang-2

Nesvacumab (Regeneron, Tarrytown, NY, USA) is a fully human IgG1 monoclonal antibody selectively binding Ang-2. In phase 2 studies of nAMD and DME, nesvacumab co-formulated with aflibercept failed to show beneficial effects over aflibercept in visual gains improvement [217].

AXT107 (Asclepix Therapeutics, Baltimore, MD, USA) is a peptide derived from the non-collagenous domain of collagen IV [217]. AXT107 modifies Ang-2 and promotes its conversion into the Tie2 agonist, and AXT107 also inhibits the signaling of VEGFR-2 and other receptor tyrosine kinases [217]. In the presence of AXT107 and Ang-2, α5β1 integrin is disrupted, promoting Tie2 clustering and converting Ang-2 into a Tie2 agonist [218]. Currently, AXT107 is in the preclinical phase of study [217].

#### 4.6.2. Bispecific Drug

Faricimab (faricimab-svoa; Vabysmo™), known as RG7716, (Roche, Basel, Switzerland and Genentech, South San Francisco, CA, USA), is a bispecific antibody binding both VEGF-A and Ang-2. Phase 3 trials for DME (YOSEMITE NCT03622580 and RHINE NCT03622593) showed robust vision gains and anatomical improvements in patients treated with faricimab and the personalized treatment interval was extended to 16 weeks [219]. In 2022, faricimab received its first approvals in the USA for the treatment of nAMD or DME [220].

#### 4.6.3. Targeting VE-PTP

ARP-1536 (Aerpio Therapeutics, Cincinnati, OH, USA) is a monoclonal antibody targeting VE-PTP. ARP-1536 is intravitreally administered, currently undergoing preclinical studies [161]. AKB-9778 (Aerpio Therapeutics, Cincinnati, OH, USA) is a small molecule antagonist of VE-PTP, increasing Tie2 phosphorylation. AKB-9778 is administered by subcutaneous injection. In preclinical studies, AKB-9778 reduced vascular leakage and ocular neovascularization, with synergistic effect when combined with VEGF inhibition [221]. AKB-9778 reduced macular edema more effectively when combined with monthly ranibizumab in a phase 2 study of DME [222].

### 4.7. Neuroprotection in DME Management

Since DR is also a neurovegetative disease, neuroprotection should be considered in the management of DR and DME. The neuroprotective agents include but not limited to erythropoietin (EPO), Cibinetide (known as ARA 290 and helix B surface peptide (HBSP)), somatostatin and brimonidine.

EPO’s protective mechanisms comprise anti-apoptosis and neuroprotection via activating the ERK and AKT pathways [40,223], neurotrophic effect and anti-reactive gliosis [224], anti-VEGF via inhibition of HIF-1α [225], anti-inflammatory effect by decreasing inflammatory factors from Müller glia [226], increase in the expression of zinc transporter 8 (ZnT8) [227], downregulation of glutamate [228], and maintenance of VE-cadherin expression via inhibiting VEGF/VEGFR-2/Src pathway. In addition, EPO is able to improve the integrity of the inner BRB [229], and maintain outer BRB integrity through downregulation of HIF-1α and c-Jun N-terminal kinase (JNK) signaling, and upregulation of ZO-1 and occludin expressions in RPE cells [39]. Recently, we found that EPO protects the inner BRB by inhibiting microglia phagocytosis via Src/Akt/cofilin signaling in experimental DR [70]. A clinical cohort study showed that intravitreal EPO could improve the visual acuity and reduce macular edema in refractory DME patients [230], demonstrating its potential usage in treatment of DME. 

Cibinetide is a synthetic 11 amino acid peptide, derived from EPO, having anti-apoptotic, anti-permeability and anti-inflammatory functions, with no erythropoietic function [231,232,233]. Both somatostatin and brimonidine were tested in diabetic patients, however, no neuroprotective effect was found for both drugs to achieve the primary endpoint [234]. However, the topical administration of somatostatin and brimonidine appears to be useful in preventing the worsening of preexisting retinal dysfunction [234]. Topical treatment with either somatostatin or brimonidine was observed to cause retinal arteriolar and venous dilation in patients with T2DM and early DR [235]. 

### 4.8. Antioxidative Therapy

Oxidative stress, resulting from the metabolic abnormalities, is regarded as a pivotal contributor to the pathogenesis of DR [236]. The reduced form of nicotinamide adenine dinucleotide phosphate (NADPH) oxidase (Nox) system is as a key enzymatic source of oxidative stress [236,237]. Nox-derived ROS contributes to retinal damage through inducing the expressions of pro-angiogenic and pro-inflammatory cytokines, including VEGF-A, EPO, Ang-2 and ICAM-1 [238,239]. The *Nox2* gene knockout reduced oxidative stress, attenuated vascular permeability, and reduced leucocyte-endothelial interaction and leukostasis in diabetic mice [237]. *Nox4* knockdown with small interfering RNA significantly decreased retinal vascular permeability, indicating the causal role of Nox4 in BRB breakdown [240]. Thus, the inhibition of Nox would provide a potential strategy for the treatment of DR and DME.

Idebenone, a ubiquinone short-chain synthetic analog, is believed to restore mitochondrial ATP synthesis with antioxidant properties [241]. Punicalagin (2,3-hexahydroxydiphenoyl-gallagyl-D-glucose), a polyphenol extracted from pomegranate (Punica granatum), is a potent antioxidant in several cell types [242]. Previous studies showed that both idebenone and punicalagin could protected RPE from oxidative damage, suggesting their possible roles in DR and DME treatment. Idebenone protected RPE through modulation of the intrinsic mitochondrial pathway of apoptosis [241]. Punicalagin exerted its effect to reduce oxidative stress and decrease the apoptosis via enhancing mitochondrial functions [242]. 

Other potential antioxidants, such as quercetin, resveratrol, curcumin, lutein, vitamin E, nicanartine, and lipoic acid, etc., are promising against oxidative stress in treatment of DR and DME [236], deserving further exploration.

### 4.9. Combo Therapy and Other Strategies

Based on the severity of the DME, combo therapy can be proposed, such as ranibizumab + OPT + 302, aflibercept/conbercept + OPT + 302 or anti-VEGF + anti-inflammatory treatment. Moreover, other approaches are also attempted to treat DME, including targeted laser photocoagulation for non-perfusion area, micropulse laser for macular microaneurysms, photobiomodulation to enhance RPE function, vitrectomy to relieve the abnormalities of vitreoretinal interface and clear vitreous body, and etc. The growing achievements of translational research will lead to future treatments for DME with better efficacy, longer duration, and greater cost-effectiveness.

## 5. Conclusions and Future Directions

DR is the leading cause of vision loss and blindness in the working-age adults with increasing incidence. DME is the main cause of vision impairment in DR patients with its pathogenesis still not completely elucidated. Although anti-VEGF therapy becomes the first-line treatment for DME, not all patients respond satisfactorily to anti-VEGF injections. The limitations and unsatisfactory response indicate other pathogenic factors, beyond VEGF, might be involved in the pathogenesis of DME (Figure 1). The advancement of the elucidation of the pathogenesis of DR and DME, including the proteomics and metabolomics studies have greatly revolutionized our recognition of DR and DME, which laid down a solid foundation for uncovering the key pathways, novel biomarkers, and establishing therapeutic targets. Based on the underlying mechanisms of DME, the future direction of DME treatment (Figure 1) should be focused on the emerging anti-VEGF agents, bispecific antibody targeting VEGF-A and other potential molecule such as Ang-2 (faricimab), targeting VEGF/VEGFR system, enhancing anti-inflammatory effect, including the suprachoroidal route delivery, integrin antagonists, NASIDs, etc. Furthermore, neuroprotection, anti-oxidation, and other approaches, such as targeted laser photocoagulation, subthreshold micropulse laser, photobiomodulation, and vitrectomy, should be considered during DME management (Figure 1). Therefore, DME treatment should be tailored treatment based on its full elucidation of the complex mechanism, which leads to the identification of new targets and therapeutic strategies for DME treatment.

## Figures and Tables

**Figure 1 cells-11-03362-f001:**
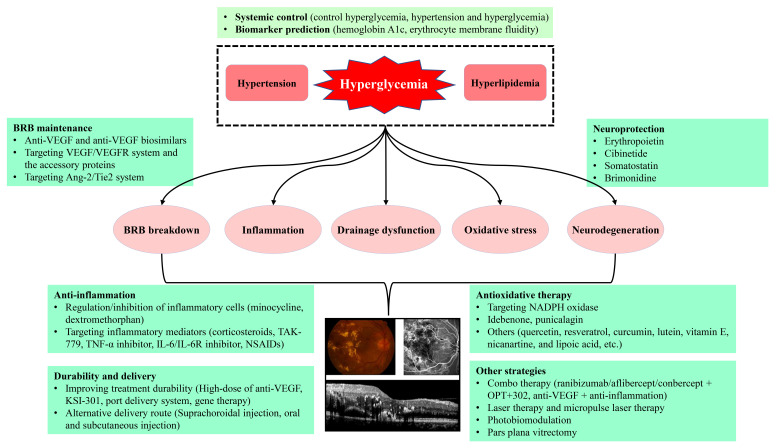
The proposed diagram for the pathogenesis and treatments of DR and DME. Anti-VEGF, anti-vascular endothelial growth factor; Ang-2, angiopoietin 2; DME, diabetic macular edema; DR, diabetic retinopathy; IL-6, interleukin 6; IL-6R, interleukin 6 receptor; NADPH, the reduced form of nicotinamide adenine dinucleotide phosphate; NSAIDs, nonsteroidal anti-inflammatory drugs; Tie2, tyrosine kinase with immunoglobulin-like and epidermal growth factor-like domains 2; TNF-α, tumor necrosis factor α; VEGF, vascular endothelial growth factor; VEGFR, VEGF receptor.

## Data Availability

Not applicable.

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
