# Peer review of "Diabetic Macular Edema: Current Understanding, Molecular Mechanisms and Therapeutic Implications"

_cells, 2022, doi:10.3390/cells11213362_

Round 1

Reviewer 1 Report

The review is very interesting and thorough.

The author described OCT-based DME classification as a single paragraph.

I suggest to add a single short paragraph for DME and OCTA as well, with the following works (Toto L, et al, Transl Vis Sci Technol. 2020 , doi:10.1167/tvst.9.7.13; Sun Z et al, Ophthalmology 2019, doi: 10.1016/j.ophtha.2019.06.016).

On page 19, Line 944: personized treatment should be replaced by tailored treatment

Reviewer 2 Report

In this review, Zhang et al. summarize the current knowledge, molecular mechanisms, and therapeutic approaches to diabetic macular edema. Overall, the work is well-structured, easy to read, and supported by good literature.

However, there are some minor points that deserve clarification and proper correction:

- line 58-59: move respectively at the end of the period ("...in the diabetic population in general, in female, and in male, respectively")

- line 63-64: change "when fluid accumulates in the macular area, it leads to..." with "fluid accumulation in the macular area leads to..."

- line 70-71: "in high-income countries, respectively"

- line 97: change "for the evaluating" with "for evaluating"

- line 103: change ; with ,

- line 128-129: change "who still exist persistent edema" with "who still exist with persistent edema"

- line 147: remove the comma after disturbance

- line 193-195: the authors mention high-resolution multi-modal imaging. Can they provide a comparison with other published works that have already used this approach for evaluating and characterizing DR and other metabolic diseases? (cfr. with metabolic imaging papers from Maulucci et al.)

- in Section 3.3, from line 228 to the end of the paragraph, the authors present some observations regarding outer BRB breakdown as per in-vitro model experiments. Since they mention cellular membrane permeability, I suggest including works and analyses on membrane fluidity of different cells (including RBC) for evaluating DR and other diabetes-related complications (see papers from Bianchetti et al.). This comparison can be included also in Section 4.1, where control of systemic risk factors is described as the first "therapeutic" approach. 

- Among the therapeutic strategies, in particular the ones related to inflammation and oxidative stress (Section 4.5), the authors should include the preventive approaches presented by Clementi et al. in their papers, including for example Idebenone and others (see 10.3390/biomedicines10020503, 10.3390/antiox11061072, 10.3390/antiox10020192)

Reviewer 3 Report

1.therapeutic strategy occupies more than half of the paper, would you please curtail it to one third or less?

2."at least 4 injections whithin 24 weeks  there are 5 or 6 months in  24 weeks, if the VEGF is injected every 4 weeks, there at least 5 or 6 times. This is also conformed to the clinical practice. 

3.in the content of introduction of inflammatory mediators, are there any up or down-regulated factors included both?

4.line 84-85,it is better to indicated clearly the difference between the two。

5.line 88-89,  pay a attention to the grammar.

6. line 168   OCT?or OCTA?reference 2 is a review, would you please tell clearly the exact original paper which observed this kind of phenomenon?

7. line 186: is or was

8. line 193 would you please give us the direct or exact evidence for iintracellular edema of RPE cells?

9.line 231 and 239:  is and can are used by wrong tense?

Round 2

Reviewer 3 Report

The  review summarizes widely the current understanding of DME including the molecular mechanisms, the changes in various factores such as proteomics,metabolomics et al, the current treatments available and  potential therapeutic implications are listed broadly. it is an excellent reference in the DR field.  however, if possible it is better to show us a diagram for them to elucidate their relationships. 
